# Statistical Power Analysis for Designing Bulk, Single-Cell, and Spatial Transcriptomics Experiments: Review, Tutorial, and Perspectives

**DOI:** 10.3390/biom13020221

**Published:** 2023-01-24

**Authors:** Hyeongseon Jeon, Juan Xie, Yeseul Jeon, Kyeong Joo Jung, Arkobrato Gupta, Won Chang, Dongjun Chung

**Affiliations:** 1Department of Biomedical Informatics, The Ohio State University, Columbus, OH 43210, USA; 2Pelotonia Institute for Immuno-Oncology, The James Comprehensive Cancer Center, The Ohio State University, Columbus, OH 43210, USA; 3The Interdisciplinary Ph.D. Program in Biostatistics, The Ohio State University, Columbus, OH 43210, USA; 4Department of Statistics and Data Science, Yonsei University, Seoul 03722, Republic of Korea; 5Department of Applied Statistics, Yonsei University, Seoul 03722, Republic of Korea; 6Department of Computer Science and Engineering, The Ohio State University, Columbus, OH 43210, USA; 7Division of Statistics and Data Science, University of Cincinnati, Cincinnati, OH 45221, USA

**Keywords:** transcriptomics, gene expression analysis, power analysis, RNA-seq, scRNA-seq, high-throughput spatial transcriptomics

## Abstract

Gene expression profiling technologies have been used in various applications such as cancer biology. The development of gene expression profiling has expanded the scope of target discovery in transcriptomic studies, and each technology produces data with distinct characteristics. In order to guarantee biologically meaningful findings using transcriptomic experiments, it is important to consider various experimental factors in a systematic way through statistical power analysis. In this paper, we review and discuss the power analysis for three types of gene expression profiling technologies from a practical standpoint, including bulk RNA-seq, single-cell RNA-seq, and high-throughput spatial transcriptomics. Specifically, we describe the existing power analysis tools for each research objective for each of the bulk RNA-seq and scRNA-seq experiments, along with recommendations. On the other hand, since there are no power analysis tools for high-throughput spatial transcriptomics at this point, we instead investigate the factors that can influence power analysis.

## 1. Introduction

Transcriptomics refers to either gene expression profiling or the study of the transcriptome using gene expression profiling technologies, where transcriptome refers to the collection of all the ribonucleic acid (RNA) molecules expressed in a cell, cell type, or organism [1]. According to the central dogma, RNA transcripts are generated by the cellular transcription process, play a role in protein-coding, and connect the genome, proteome, and cellular phenotype [2]. Therefore, as a proxy for proteome analysis, numerous transcriptomic studies have analyzed messenger RNA (mRNA) molecules encoding proteins [3]. In addition, transcriptomic approaches have contributed to the advancement of various biological and medical studies, such as cancer biology, by identifying possible prognostic biomarkers [4].

Transcriptomic studies can be categorized using underlying gene expression profiling technology, and technological advancements have increased the scope of target discovery. Figure 1 provides a summary of three types of gene expression profiling technologies in terms of their profiling resolution, data structure, and potential target discoveries. Hong et al. [4] illustrate the evolution of RNA sequencing technology. Unlike microarrays, which profile predefined transcripts through hybridization, bulk RNA sequencing (bulk RNA-seq) allows a genome-wide analysis across the whole transcriptome within a cell population by employing next-generation sequencing (NGS) technology [5]. In contrast to bulk RNA-seq, single-cell RNA sequencing (scRNA-seq) enables the comparison of the transcriptomes of individual cells and the analysis of heterogeneity within a cell population [3]. The high-throughput spatial transcriptomics (HST) technology permits gene expression profiles at the cell or close-to-cell level while also preserving spatial tissue context information [6]. We note that the characteristics of the transcriptomic data are contingent on the underlying technology. Bulk RNA-seq data are highly reproducible, indicating that technical replicas display minimal systemic changes and are thus unnecessary [7]. Bacher and Kendziorski [8] demonstrate that scRNA-seq data has a greater proportion of zeros, more variability, and a more complex distribution than bulk RNA-seq data.

When designing a transcriptomic experiment, it is crucial to determine the experimental factors, such as the number of biological replicas, the number of cells, and the sequencing depth, to guarantee sufficient power. In the statistical framework, power refers to the probability of detecting target discoveries, also known as sensitivity. In bulk RNA-seq analysis, Schurch et al. [9] provided an empirical guideline for the number of biological replicas required to guarantee sufficient power, and Liu et al. [10] demonstrated that the number of biological replicas has a greater influence on power than the sequencing depth. Pollen et al. [11] demonstrated that low-coverage scRNA-seq is sufficient for cell-type classification. Despite the existence of basic guidelines, there exists no unifying rule due to the complexity of power. For example, biological factors of the experimental unit, such as sex and breeding type, may impact power and should be considered more systematically when selecting experimental parameters.

Therefore, to determine the experimental factors in transcriptomic experiments in a systematic way, a power analysis should be conducted. Cohen [12] pioneered the concept of power analysis, which refers to the examination of the relationship between power and all of the parameters influencing power. These parameters include the desired error rate and the size of the experimental effect of interest (effect size). In practice, power analysis aims to identify a parameter under the assumption that all other parameters remain constant, with power itself being considered a parameter. In power analysis, sample size or power itself is a common target parameter [13]. In this review paper, the sample size refers to either the number of biological replicas or the number of cells. In an experimental study, power analysis provides crucial information at each stage of the experiment. Before the study, prospective power analysis helps determine the experimental factors that will provide sufficient power for detecting target discoveries. Researchers can conduct a retrospective power analysis to evaluate the experiment, despite differing opinions regarding how to use the collected data for the power analysis, as discussed in Thomas [14].

Power analysis varies according to the underlying objectives of the study and how the data will be analyzed to achieve the research objective [15]. As previously discussed, the technology employed affects the scope of target discoveries and the transcriptomic data characteristics. In this context, the power analysis for three distinct transcriptomic technologies will be examined, including the bulk RNA-seq, scRNA-seq, and HST technologies. In Section 2, Section 3 and Section 4, each transcriptomic technology is covered in a separate section. For a given technology, we examine the power analysis for transcriptomic experiments with respect to experimental factors, research objectives, and explanations of the existing power analysis tools. If there are power analysis tools for a particular technology and research objective, we provide recommendations, while tutorials are also provided for some of these recommended tools in the Appendix A.

## 2. Power Analysis for Bulk RNA-Seq Experiments

### 2.1. Bulk RNA-Seq Experiment

Sequencing technologies originate from Sanger sequencing, first introduced by Sanger et al. [16]. In 2005, the introduction of Next-Generation Sequencing (NGS), also known as massively parallel sequencing, improved sequencing in terms of high throughput, scalability, and speed. In particular, NGS technology enables the bulk RNA-seq profiling of gene expression levels in over ten thousand genes simultaneously in a specific tissue or cell population, where the gene expression is characterized by an abundance of messenger RNA (mRNA). The typical bulk RNA-seq protocol includes sample preparation, mRNA fragmentation, reverse transcription to complementary DNA (cDNA), and the mapping of cDNA fragments to a reference genome. A gene’s expression level is ultimately determined by counting the cDNA fragments, called reads, that are mapped to the gene. See Stark et al. [17] and Van den Berge et al. [18] for more details. Sequencing depth is defined as the total number of reads, influencing the sequencing’s technical precision [19]. The bulk RNA-seq profiling platforms include Illumina’s HiSeq and MiSeq and ABI’s SOLID. Hong et al. [4] illustrate the technological evolution over time of RNA sequencing and provide in-depth explanations of the related platforms.

Bulk RNA-seq transcriptomic experiments typically aim to identify differentially expressed genes (DEGs) across various experimental conditions, where multiple biological replicas are expected in each condition. DEGs are the bulk RNA-seq experiment’s detection target, with their detection probability determining the associated power. Specifically, the power of the bulk RNA-seq gene expression analysis is defined by the expected proportion of DEGs detected among all DEGs, following a prespecified statistical procedure. Unlike conventional microarray technology that generates continuous data, bulk RNA-seq generates count data. Due to the discrete nature of the data, the Poisson distribution was originally employed to model the bulk RNA-seq data. However, due to its one-parameter nature, the Poisson distribution cannot account for extra-biological variation in bulk RNA-seq data. Therefore, the negative binomial (NB) distribution, which can be viewed as a Poisson-gamma mixture, has gained popularity. Under a model assumption, a DEG is characterized as a gene whose mean expression ratio (i.e., fold change) deviates from one for any pair of experimental conditions. The difference or ratio can be understood as a measure of the effect size that characterizes the DEGs. The bioconductor packages of ‘edgeR’ [20], ‘DESeq’ [21], ‘DESeq2′ [22], and ‘baySeq’ [23] employ the NB model to identify DEGs. In addition, Van den Berge et al. [24] further improved these approaches by considering zero-inflated gene expression using a zero-inflated NB model. While NB-based methods generally have a higher detection power, there are also reports indicating its false discovery rate (FDR) inflation [25,26] due to ignoring the uncertainty of the estimated dispersion parameters [27]. Alternatively, the voom method [28] can be used to detect DEGs by applying normal-based theory to the log-transformed count data, which is implemented in the limma Bioconductor package. Even though count data is not directly modeled, the voom method adjusts heterogeneous variances across all observations concurrently by utilizing an adequate mean and variance relationship. Additional software tools for DEG analysis are described in Schurch et al. [9] and Stark et al. [17].

In the case of a bulk RNA-seq experiment, it is essential to determine the number of biological replicas that will provide sufficient DEG detection power, a type of power analysis. Consider the factors that may affect the power. Note that the power depends on the assumed model’s parameters and the software tools that provide the *p*-value for each gene under consideration. Additionally, the power is affected by the considered error rate and the target level. Bulk RNA-seq gene expression analysis typically considers multiple genes. When multiple genes are simultaneously inferred, it is common to control the FDR rather than the type 1 error rate, which is appropriate for inferring a single gene. By controlling FDR, it is possible to regulate the proportion of non-DEGs among the genes declared to be DEGs on average. Consequently, when inferring multiple genes and conducting a power analysis, it is necessary to consider the target FDR level.

### 2.2. Bulk RNA-Seq Power Analysis Tools

Numerous power analysis software tools calculating the number of biological replicas, alternatively the sample size, for bulk RNA-seq experiments have been developed according to the factors affecting the power: the model assumptions, the testing type employed for each gene, and the desired error rates to be controlled. Model parameters are often estimated using pilot data, and some tools provide stored data for this purpose. As demonstrated by data analysis in Poplawski and Binder [29], if the stored data are utilized carelessly, a highly inappropriate sample size can be suggested. In addition to sample size, some software tools consider sequencing depth to be an experimental factor that influences the power to be chosen during experimental design. Liu et al. [10] demonstrated the trade-off between biological replicas and sequencing depth in the context of statistical power.

Hart et al. [19] suggested a flexible power analysis approach that calculates the sample size for a single gene expression analysis using the NB model, which is implemented in the ‘RNASeqPower’ Bioconductor package. Due to the asymptotic normality of the score test statistic, a closed-form power function is obtained as a function of all possible parameters, including sample size, fold change, average sequencing depth, target type 1 error rate, and coefficient of variation. Due to the simplicity of the inference situation and the closed-form power function, it is possible to perceive the relationship among all of the parameters affecting the detection power. Hart et al. [19] also suggested a sequencing depth motivated by the parameters’ relationship and demonstrated that, although the method does not assume FDR control, it can be extended to multiple gene inference by setting the *p*-value threshold α to a small value, such as 0.001.

Li et al. [30] proposed a tool for calculating sample size based on the NB model and FDR control via a gene-specific power function. The approach is effectively implemented in the ‘RnaSeqSampleSize’ Bioconductor package, with an additional parameter estimation procedure supported by data. However, the ‘RnaSeqSampleSize’ tool tends to overestimate the sample size in the data analysis and data-based simulation study of Poplawski and Binder [29]. To overcome this overestimation, Bi and Liu [31] suggested a method that assumes the NB model but uses the normal-based test statistic via the voom method to assess the power function partially analytically, implemented in the ‘ssizeRNA’ R package. According to the data-driven simulation study of Poplawski and Binder [29], this approach is faster and provides the sample size closer to the actual number required to achieve the desired power, compared to other approaches. Additionally, Wu et al. [32] proposed a simulation-based FDR controlling approach, implemented in the ‘PROPER’ tool. Table 1 provides a summary of the information from different power analysis tools. The tools are chosen from the methods with relevant literature described in Poplawski and Binder [29].

### 2.3. Bulk RNA-Seq Power Analysis Tool Recommendation

The ‘ssizeRNA’ R package was chosen based on the outcomes of two simulation studies by Poplawski and Binder [29] and Bi and Liu [31]. From the six power analysis tools mentioned in Table 1, we first considered ‘RnaSeqSampleSize’, ‘ssizeRNA’, and ‘PROPER’ based on their FDR-targeting nature and focus on a single DEG analysis tool. However, based on its performance in the simulation studies, we decided to exclude ‘RnaSeqSampleSize’ from consideration. Specifically, according to Poplawski and Binder [29], ‘RnaSeqSampleSize’ typically recommends a very large sample size. ‘RnaSeqSampleSize’ performs well in Bi and Liu [31] when the model is simple and gene-specific parameters are absent. When the simulation model became realistic, the sample size suggested by ‘RnaSeqSampleSize’ was either too large to significantly exceed the desired power or too small to adequately regulate power. The subsequent selection was based on speed. The simulation results presented in both papers indicate that both the ‘PROPER’ and ‘ssizeRNA’ tools recommend sample sizes with target power levels. Due to the conservative nature of the voom method, the ‘ssizeRNA’ tool typically recommends a few more samples. In terms of usability, however, we recommend the ‘ssizeRNA’ tool, which is faster due to its analytical nature. Based on this rationale, in this paper, we provide a comprehensive tutorial for ‘ssizeRNA’ in Appendix A.

## 3. Power Analysis for Single-Cell RNA-Seq (scRNA-Seq) Experiments

scRNA-seq technologies have revolutionized the study of transcriptomics by profiling genome-wide gene expression at the individual cell level. The cell-level information provides unprecedented opportunities for studying cellular heterogeneity and expands our understanding of developmental biology [3]. Even though the context of a single-cell transcriptomic study differs from that of a bulk transcriptomic study, DEG detection remains a fascinating study area. In addition, the cell information enables researchers to answer questions about cell subpopulations. The relevant power analysis has been developed in response to the distinct research questions. In general, the sample size in scRNA-seq experiments refers to the number of cells. Due to the additional technical steps required to distinguish cells, scRNA-seq data contain more zeros than bulk RNA-seq data [11], and a zero-inflated model is frequently employed when developing statistical approaches [36]. Section 3.1 and Section 3.2 discuss a power analysis for identifying cell subpopulations and detecting DEGs, respectively, for scRNA-seq experiments. The information presented in Table 2 outlines a variety of power analysis tools applicable to single-cell transcriptomic experiments with distinct research questions.

### 3.1. Power Analysis for Cell Subpopulation Detection

Unlike bulk RNA-seq experiments, scRNA-seq experiments frequently attempt to identify the characteristics underlying cell subpopulations. A cell subpopulation refers to a group of cells determined by various cell types, states, or subclones. Bulk RNA-seq data does not allow cell subpopulation-level investigation, especially for rare cell subpopulations. In contrast, the scRNA-seq data provides the cell subpopulation-level resolution [45]. The research questions and associated power analysis can be further divided into two categories, depending on whether scRNA-seq experiments examine the proportion of cell subpopulations within a single tissue (Section 3.1.1) or the proportional differences across experimental conditions for a given cell subpopulation (Section 3.1.2).

#### 3.1.1. Ascertaining Cell Subpopulation Proportions in a Single Tissue

Multiple cell types in varying proportions compose a biological tissue. In the experimental design phase, power analysis is indispensable for ensuring that enough cells are sampled to adequately represent both normal and rare cell types. The following sections discuss the power analysis for sufficient cell numbers (sample size) in a single tissue.

Two software tools, ‘howmanycells’ (https://satijalab.org/howmanycells accessed on 25 December 2022) and ‘SCOPIT’ [37], were developed specifically for cell number calculation. Using statistical models, they both approached the problem by calculating the probability of sampling at least a predetermined number of cells from each subpopulation. The ‘howmanycells’ function uses the NB distribution to estimate the total number of cells required for adequate representation of a given cell subpopulation under the assumption that the number of cells of each cell type is statistically independent. This assumption may not hold in practice, but the results can be used to determine the required minimum sample size. On the other hand, ‘SCOPIT’ employs the Dirichlet-multinomial model for the distribution of the number of cells from each subpopulation, which more accurately reflects the constraint on the proportion of cell subpopulation (i.e., proportions sum to one).

Both ‘howmanycells’ and ‘SCOPIT’ are comparable in that they use analytical approaches, identify the proportion of the rarest cell type as the most significant statistical factor affecting power, and offer lightweight web applications to facilitate quick and intuitive power calculation. Above all, their estimates of the required sample size are comparable in general. An important distinction is that ‘SCOPIT’ permits retrospective analysis for hypothetical experiments, i.e., determining how many cells would be required based on the number of sequenced cells, the number of subpopulations detected, and their frequencies. In addition, ‘SCOPIT’ reports Bayesian credible intervals for the estimated probability and number of cells to account for the uncertainty associated with the observed empirical subpopulation frequencies.

The methods mentioned above only consider the effects of cell subpopulation proportions and total cell number and do not account for technical factors such as sequencing depth. This is partially due to the difficulties in obtaining an analytical solution when other factors are considered. Note that these methods are intended to estimate the total number of cells in a single biological sample to identify subpopulations. When dealing with multiple samples in scRNA-seq experiments, the detection target may change, and different approaches are needed. The following section describes the problem.

#### 3.1.2. Ascertaining Differential Cell Subpopulation Proportions between Distinct Experimental Conditions

In a multi-sample scRNA-seq experiment, researchers are primarily interested in determining whether a specific cell subpopulation has differential abundances between experimental conditions (e.g., diseased vs. healthy). In this case, the difference in cell subpopulation proportion represents the effect size, and the number of biological samples (such as patients and mice) represents the sample size. The proportion of the cell subpopulation, the number of cells, and the number of (biological) samples influence the power. Since cell subpopulations are often identified by comparing marker gene expression levels, sequencing depth may affect power, as it influences technical variation. Moreover, since this is a multiple-sample experiment, the batch effect may be significant, and the experimental design may be unbalanced. Consequently, the batch effect and experimental design (balanced or unbalanced, paired or unpaired) may also impact the power.

Two approaches, ‘Sensei‘ [38] and ‘scPOST’ [39], have been developed for the power analysis of distinguishing proportional differences within a cell subpopulation. The former provides an analytical solution after a reasonable approximation, whereas the latter relies on simulation. They both consider the potential impact of the proportion of the cell subpopulation (biological factor), the number of cells, and the number of samples (experimental factors), but only ‘scPOST’ considers the effect of gene expression variation. Both approaches attempt to explain how to balance the number of biological samples and the number of cells within a limited budget, and both suggest that increasing the sample size yields greater power than increasing the number of cells per sample. In addition, ‘scPOST’ indicates that modest reductions in sequencing depth have negligible effects on power.

Specifically, ‘Sensei’ integrates the impacts of the number of cells and the number of biological replicas in a mathematical framework. It models the abundance of cell types using a beta-binomial distribution and estimates the sample size based on Welch’s t-test. Under this framework, beta distribution captures the biological difference in cell type abundance between groups, as well as the variance among samples within a group, while binomial distribution models the technical variation caused by a limited number of cells. ‘Sensei’ provides a closed-form representation for the statistical power upon reasonable approximation, which makes a lightweight web application possible. As an output, ‘Sensei’ shows a table of false negative rates for each feasible sample size combination.

Although ‘Sensei’ attempted to account for some biological and technical variations, the pursuit of an analytical representation of power necessitates the adoption of assumptions and simplifications that may not apply to real data (e.g., assume no batch effect). In contrast, ‘scPOST’ employs a simulation-based method to account for the effects of more factors. It begins by estimating key parameters based on the prototype or pilot data supplied by the user. Specifically, it assumes gene expression variation in the principal components (PCs) space that arises from three sources (batch, sample, and residual), and employs linear mixed effects models to decompose the total variance for each PC and each cluster. Both fixed and random effects are extracted from the fitted models, and the cluster frequency mean and covariance are estimated from the prototype dataset. In the second step, the previously estimated parameters and user-specified batch and sample effect scale parameters are used in linear mixed effects models to simulate PC coordinates for cells. In the final step, ‘scPOST’ employs a test based on logistic mixed effects models to determine whether the mean frequency of a cluster differs significantly between two conditions. The power is computed as the proportion of simulation runs in which at least one cluster represented differential abundance.

### 3.2. Power Analysis for DEG Detection

Identifying DEGs is another important goal of scRNA-seq data analysis. DEG analysis can also be divided into two categories, depending on whether the goal is to identify (i) DEGs across different conditions (e.g., treatment vs. control) for a specific cell type or (ii) DEGs that are differentially expressed across cell types for a given biological sample. Numerous factors can influence power, such as effect size, number of cells, number of biological replicas, sequencing depth, dropout rates, cell subpopulation proportion, and multiple testing methods. Given that so many factors may affect power, it is hard to provide an analytical framework to assess power. Therefore, most of the existing work employs simulation-based approaches, which consist of three key steps: parameter estimation, data simulation, and power evaluation. In the parameter estimation step, important parameters like the gene-wise mean and standard deviation are estimated from user-provided data or representative example data based on a data model. In the simulation step, gene expression values are simulated based on the estimated parameters. Finally, in the power evaluation step, existing DEG analysis or detection methods are applied to the simulated data to assess power. The subsequent sections discuss the approaches in detail.

#### 3.2.1. DEGs across Different Conditions for a Cell Type

Similar to bulk RNA-seq experiments, a DEG analysis can be performed to identify genes whose expression levels vary significantly between experimental conditions. In scRNA-seq experiments, such DEG analysis is often performed for a specific cell type. Four software tools are available for this type of power analysis: ‘powsimR’ [42], ‘hierarchicell’ [41], ‘POWSC’ [43], and ‘scPower’ [40]. ‘powsimR’ and ‘POWSC’ are more suitable for single-sample experiments, while ‘hierarchicell’ and ‘scPower’ are designed for multi-sample experiments. ‘powsimR’ assumes an NB distribution for the count data and emphasizes the mean-dispersion relationship during simulation. The existing package is used for DEG detection, and power-related statistics including the FDR and true positive rate (TPR) are calculated to evaluate power based on the estimated and simulated expression differences. The ‘hierarchicell’ also assumes an NB distribution for the gene expression value, and it highlights the hierarchical structure of scRNA-seq data from multiple individuals. For power evaluation, it implements a two-part hurdle model.

‘scPower’ uses an analytical-based approach for this task. The fundamental idea behind ‘scPower’ is that a gene needs to be expressed and to exceed a significance cutoff to be identified as a DEG. Therefore, it decomposes the power as the product of the expression probability (probability of detecting an expressed gene) and the DE power (probability of being significantly expressed). For the expression probability, a pseudobulk approach is adopted. Specifically, it sums the expression of a gene over all cells of the cell type of interest within an individual to obtain the pseudobulk count for that gene. Then it calculates the probability of this pseudobulk count being greater than a threshold based on an NB distribution. Based on this probability, the probability that the gene is expressed is obtained from a cumulative binomial distribution. The DE power is calculated analytically based on an NB model using existing tools.

#### 3.2.2. DEGs across Different Cell Types

Identifying genes that are differentially expressed across different cell types under the same experimental condition is another common DEG analysis, aiming to identify genes that could distinguish from one cell type to another. ‘scDesign’ [44] and ‘POWSC’ [43] were developed for the power analysis, and both are simulation-based approaches designed for studies involving a single biological sample. ‘scDesign’ assumes a gamma-normal distribution for log-transformed count data. ‘POWSC’ assumes a mixture of zero-inflated Poisson and lognormal-Poisson distributions for the count data. ‘scDesign’ and ‘POWSC’ allow user-supplied data for parameter estimation, while ‘POWSC’ also provides precalculated parameter estimates from various tissue types. The parameters to be estimated for ‘scDesign’ include the cell library size and cell-wise dropout rate, as well as the gene-wise mean, standard deviation, and dropout rate. The parameters to be estimated for ‘POWSC’ include the cell-wise zero inflation point mass and Poisson rate, as well as the gene-wise mixture proportion, mean, and variance. In the data simulation step, both approaches consider the constraint on total reads and allow users to choose the number of cells and the sequencing depths under the constraint. Therefore, they can provide insights regarding how to optimize the trade-offs between these two experimental factors. ‘scDesign’ performs DEG analysis using a two-sample t-test and reports five power-related measures. On the other hand, ‘POWSC’ utilizes existing DEG analysis tools and reports both stratified and marginal power.

### 3.3. scRNA-Seq Power Analysis Tool Recommendations

As illustrated in Table 2, for the scRNA-seq experiments, a unique set of software tools for power analysis has been developed for a specific research objective. Specifically, the tools’ distinctive features include the factors considered and the data models. Therefore, users should consider the previously stated distinctive features when selecting an appropriate power analysis tool. Here, we make recommendations based on these considerations.

First, the ‘SCOPIT’ tool is recommended when detecting cell subpopulations is the purpose of the research. In this case, one can choose between ‘howmanycells’ and ‘SCOPIT’. Both offer lightweight web applications to facilitate fast and intuitive power calculations, and their estimates for the required number of cells are nearly identical. However, we recommend ‘SCOPIT’ for this research purpose given its more comprehensive and kinder documentation.

Second, when the differential proportion of cell subpopulations is the main goal of the research, one can choose between ‘Sensei’ and ‘scPOST’. ‘Sensei’ provides a lightweight web application that is quick and intuitive. However, ‘scPOST’ allows the consideration of more factors because it is a simulation-based method. If users desire a quick and approximate estimate of the number of cells, ‘Sensei’ is a suitable option. On the other hand, ‘scPOST’ may be preferred if users wish to consider various experimental and biological factors, such as the batch effect and gene expression variation, in the statistical power analysis.

Third, ‘scPower’ and ‘hierarchicell’ are available tools for power analysis if researchers wish to identify the genes whose expression levels differ under different experimental conditions within a particular cell type, and multiple biological samples are involved. Between these two tools, we recommend ‘scPower’ over ‘hierarchicell’ due to its user-friendly web application. Likewise, ‘POWSC’ and ‘powsimR’ can accomplish the task with a single sample. Between these two tools, we recommend ‘POWSC’ over ‘powsimR’ because of the richer documentation for ‘POWSC’.

Finally, if the genes differentiating one cell type from another are the primary objective, then ‘scDesign’ and ‘POWSC’ can assist. They address the restriction on total sequencing depth and the zero-inflation issue, although they employ different data models. Between these two tools, we recommend ‘POWSC’ over ‘scDesign’ because ‘POWSC’ also reports the stratified power, i.e., stratified based on gene expression level or zero fractions, which makes more sense given that power depends on these two factors.

In this paper, we provide a comprehensive tutorial for ‘POWSC’ in Appendix A. We do not provide tutorials for other recommended tools because using ‘scPower’ and ‘SCOPIT’ are intuitive enough on their own, while ‘scPOST’ is already accompanied by a comprehensive and straightforward vignette.

## 4. Power analysis for Spatial Transcriptomic Experiments

### 4.1. Introduction of High-Throughput Spatial Transcriptomics (HST) Technology

The lack of spatial information has limited the scope of scRNA-seq data analysis. Technological advancements in HST have made it possible to collect gene expression data along with spatial coordinates. HST technology enables gene expression profiling while preserving the spatial location (coordinate) of each observational unit, as depicted in Figure 1. The observational unit can be a cell or a group of cells (spot). There are two main categories of technological variations of HST technology: imaging-based and sequencing-based. ‘seqFISH+ [46] and MERFISH [47] are representative technologies for generating imaging-based HST data with a cell as the observational unit. Due to its probe hybridization-based gene detection, imaging-based HST data can only observe a limited number of genes. A standard technology for generating sequencing-based HST data with a spot as the observational unit is 10X Visium [48]. Since sequencing-based technology employs NGS technology, there are fewer restrictions on the number of genes compared to imaging-based technology. Accordingly, there is currently a technological trade-off between the cell resolution and the number (dimension) of genes. For instance, imaging-based HST data can be described as high-resolution and low-dimensional data, while sequencing-based HST data can be considered to be low-resolution and high-dimensional data. Note that the spatial information from various HST data types is derived from distinct observational units (cells and spots), which affects the type of inferences we can make. For example, image-based HST data would be more suitable for statistical inferences requiring cell-level resolution.

As illustrated in Figure 2, researchers can answer multiple research questions using the HST data, including spatially variable gene (SVG) detection, tissue architecture identification, and cell-cell communication prediction. Answering these research questions requires understanding how to incorporate spatial information into a model to define the SVGs, tissue architecture, and cellular phenotype. First, the SVG detection method determines which genes exhibit spatial patterns within the target tissue. Observing SVGs reflects their biological function and is often the first analysis conducted, and examples include ‘spatialDE’ [49], ‘SPARK’ [50], ‘Trendsceek’ [51], ‘BOOST-GP’ [52], ‘BOOST-MI’ [53], and ‘BinSpect’ [54]. ‘spatialDE’ and ‘SPARK’ utilize the Gaussian random effect model and the Poisson log-normal model, respectively, with distinct normalization strategies. On the other hand, the ‘Trendsceek’ approach detects spatial variation using a nonparametric approach. ‘BOOST-GP’ and ‘BOOST-MI’ are based on Bayesian frameworks, where ‘BOOST-GP’ focuses on handling the zero-inflation while ‘BOOST-MI’ considers the lattice data structure. Lastly, ‘BinSpect’ is a computationally efficient approach that employs the statistical enrichment of spatial network neighbors utilizing binarized expression data, where the binarization is based on either k-means clustering (k = 2) or rank thresholding. Second, the main goal of tissue architecture identification is to group the observational units (i.e., cells or spots) into biologically distinct clusters. Before the advent of HST technologies, previous studies employed clustering based only on the gene expression data [55,56]. Now, additional spatial information available in the HST data allows one to also consider the proximity between cells to improve such clustering. ‘HMRF’ [54], ‘BayesSpace’ [57], and ‘SPRUCE’ [58] are examples of models employing the spatial associations between observational units to identify clustering patterns. Third, cell-cell communication analysis is used to predict interactions between cells. The spatial closeness or adjacency can provide important information to improve this type of analysis because spatially closer cells are more likely to interact with each other. Previously, in the absence of spatial information, interactions between ligands and receptors were predicted only based on their gene expression patterns [59,60]. For example, ‘CellChat’ [61] estimates the interaction between ligands and receptors based on the latent distance between cells, which is calculated solely based on gene expression data. This does not reflect the fact that cells located nearby are more likely to interact with each other; incorporating such information can lead to higher accuracy.

Given the coordinates from each observation in the HST data, the spatial patterns are modeled through the distances among observations. We note that the optimal approach to calculating the distances among observations can be different for different data types. Figure 3 illustrates how the imaging-based and sequence-based HST data can be regarded as different types of spatial data. First, one can consider the imaging-based HST data as spatial marked point process data. Here, geostatistical data follows a spatial process that varies continuously, but is observed only at discrete points (coordinates). By using the coordinate information, we can define the distance (e.g., Euclidean distance) among cells. The existing models, including ‘spatialDE’ and ‘SPARK’, define the spatial closeness by calculating the distances among cell coordinates. On the other hand, the sequencing-based HST data can be thought of as lattice or areal data observed at discrete points or spots on a regular or irregular grid. In the lattice data structure, the neighborhood is defined by the adjacency on the grid and the distance between two spots is measured by the least number of spots that need to be visited while moving from one spot to the other on the lattice.

As shown in Figure 4, there are several key experimental factors that can affect the generation of spatial features in HST data, including the choice of tissue area, the size of the fields of view (FoVs), the number of FoVs, and the number of cells or spots, where FoVs are defined as the region on a tissue captured by an HST experiment. Note that such selection of FoVs and tissue area is needed as it is often not possible to capture the whole tissue using the HST experiment. These experimental factors can affect the capture of transcripts at a specific location on a tissue [62] or lead to a different context for capturing the region of interest, e.g., building a neighborhood network [63]. Hence, the power analysis for HST data needs to take these experimental factors into account to estimate the minimum number of samples to achieve a specific analysis goal using HST data. First, the size of the FoVs determines on what scale we measure spatial features and gene expression locally (i.e., local capture efficiency). On the other hand, the number of FoVs affects how many different regions we check on a tissue (i.e., global capture efficiency). Second, because these FoVs are not qualitatively and biologically identical, it also matters where on the tissue we capture. For example, for tissue architecture identification, one might want to include the regions that contain interesting and/or rare cell subpopulations. Likewise, for predicting cell-cell communication, one might hope that the regions with active cell-cell interactions are included in our HST data. Third, because the number of cells and spots can affect the signal-to-noise ratios of the generated HST data, one needs to make sure that sufficient cells and spots are captured to avoid potential analytical and computational issues. In summary, a rigorous experimental design that systematically considers these experimental factors will facilitate the effective use of resources (e.g., experimental cost) by improving efficiency in capturing the spatial features with gene expression data.

### 4.2. Literature Reviews of Power Analysis for HST Data

Recently, Bost et al. [64] implemented several experiments to figure out how the number of FoVs and their widths affect the coverage of the true clusters in a tissue. By changing the number and the size of FoVs, they examined the ratio of the number of covered clusters to the true number of clusters. It was the first attempt to investigate how the experimental design affects the HST data analysis. For example, they calculated the required number of FoVs to discover the true clusters in the cell phenotype and compared it between tumor samples and healthy samples. The result showed that a larger number of FoVs are needed to capture the true clusters in tumor samples compared to healthy samples, likely because of the complex and heterogeneous tissue structure generated through tumorigenesis. They also applied this experiment to real data on heart disease and breast cancer. They concluded that different types of data, such as human body and animal tissue, have different required numbers and sizes of FoVs for recovering the true clusters. Moreover, the technologies of generating the HST data also affect the relationship between the identification of cell clustering and the number and size of FoVs. However, the investigation of Bost et al. [64] is limited in the sense that it was based on an empirical equation that was not justified by any statistical model or machine learning model. Moreover, its ratio of discovering the true cluster is not the power required to discover the true clusters, whose computation requires a large number of iterations.

In contrast to Bost et al. [64], who used an empirical equation to calculate the ratio of covering true clusters, Baker et al. [65] employed a simulated HST approach to investigate the design of HST experiments. Here, they performed a spatial power analysis experiment with their devised HST data generation, called the “*in silico*” approach. Using the in silico approach, they generated various types of HST data as spatial profiling data such as cells in random states or cells in self-preference states to proceed with an exploratory computational framework. They pointed out three experimental factors to be considered in calculating the power: the number of cells, the number of FoVs, and the size of the FoVs. They applied their approach to two analytical tasks, including cell type discovery (tissue architecture identification) and cell-cell communication. Based on these simulation strategies, they used statistical models such as the Gamma-Poisson model to predict how many FoVs are required to discover the cell types or cell interactions. Through their simulation studies, they discovered that the size of the FoVs and the number of FoVs impacted the statistical power. First, in cell type discovery, they concluded that the nature of tissue structure affects the required number of cells and FoVs to discover the true cell types. They demonstrated this by applying the power analysis model to unstructured data of human breast cancer, highly ordered and heterogeneous data from the mouse brain, and complex and recurrently structured data from the mouse spleen. Second, for the cell-cell communication task, they argued that the interactions among the cells might not be captured with an insufficient FoV size. However, the investigation of Baker et al. [65] also has multiple limitations. First, it is hard to directly apply their approach to point-referenced data (point process data). Specifically, the simulation data generation model (“*in silico*”) is based on a blank tissue scaffold where the random circle packing forms a planar graph, which requires strong prior knowledge for cluster labels. This cannot capture all the variations in point reference data whose spatial locations are randomly distributed, and the resulting pattern often exhibits non-trivial microscale variation. Second, their investigation was limited to the number and sizes of the FoVs, and they ignored other important experimental factors that can affect the statistical power, e.g., the choice of tissue area and the number of cells/spots mentioned in Figure 4. In summary, at this point, the optimal strategies for statistical power analysis for HST experiments remain to be explored.

## 5. Conclusions

The advancement of transcriptomic technology has allowed researchers to expand their scope of questioning. In order to guarantee biologically meaningful findings, rigorous experimental design is critical, including a statistical power analysis that carefully considers research questions and data characteristics. In this review paper, we investigated the power analysis for three distinct types of transcriptomic technologies from a practical standpoint. First, in the case of the bulk RNA-seq experiment, the primary objective is to identify DEGs, and we recommend the R package ‘ssizeRNA’ as a tool for power analysis. Second, in the case of the scRNA-seq experiment, two main analytical goals are cell subpopulation identification and DEG detection. Regarding cell subpopulation detection, specifically, we recommend ‘SCOPIT’ for detecting cell subpopulations and ‘scPOST’ for inferring proportional differences across cell subpopulations. Regarding DEG detection, we recommend ‘scPower’ for DEG detection across multiple cell subpopulations using multiple samples, and ‘POWSC’ for DEG detection across cell subpopulations with a single sample and within a cell subpopulation under varying experimental conditions. Third, in the case of the HST experiment, its power analysis framework is still under-developed, and we highlight key aspects that need to be considered for the power analysis framework of HST experiments, including research questions (SVG, tissue architecture, cell-cell communications), technological variations (imaging- and sequencing-based HST), and experimental factors (tissue area, the number and size of FoVs, and the number of cells or spots).

We note that there are still some exciting and interesting directions that were not sufficiently discussed in this paper, especially regarding HST experiments. Specifically, in this paper, we did not discuss integrative analysis frameworks for HST data analysis, partially because the research on statistical approaches in this setting is still actively ongoing at this point. First, in this paper, we did not discuss power analysis for multi-sample HST experiments. However, the joint analysis of multi-sample HST data has great potential to increase the detection power [66,67], and this direction needs to be investigated more in the future. Second, the integrative analysis of HST data with corresponding pathology images also recently gained significant attention, and examples of this include ‘SpaGCN’ [68]. As such integrative analysis is expected to provide improved statistical power, this direction will deserve more in-depth review in the future. On the other hand, in this review, we tried to focus on discussing key experimental factors that can affect the statistical power. However, clearly this is not a comprehensive list of the important factors we need to consider in the design of HST experiments. For example, cell type mixture is an important issue for sequencing-based HST data analysis. Cell type deconvolution, a technique that tries to address this issue [69,70,71], and its links to statistical power should also be an interesting topic to investigate.

We believe that this review paper can be a useful guideline for the future design and statistical power analysis of transcriptomic experiments.

## Figures and Tables

**Figure 1 biomolecules-13-00221-f001:**
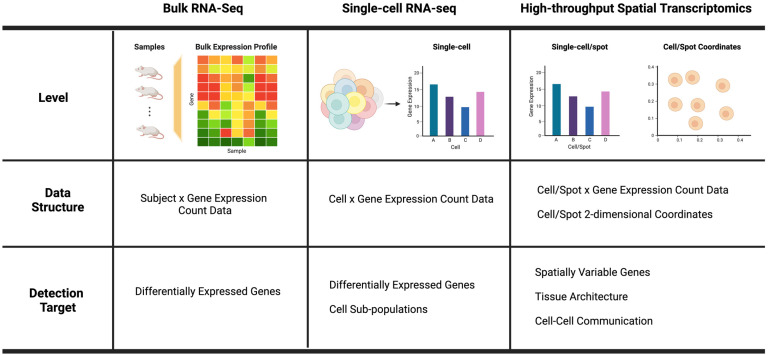
Comparison of bulk RNA-seq, single-cell RNA-seq, and high-throughput spatial transcriptomics technologies in terms of the profiling resolution (level), data structure, and target discoveries.

**Figure 2 biomolecules-13-00221-f002:**
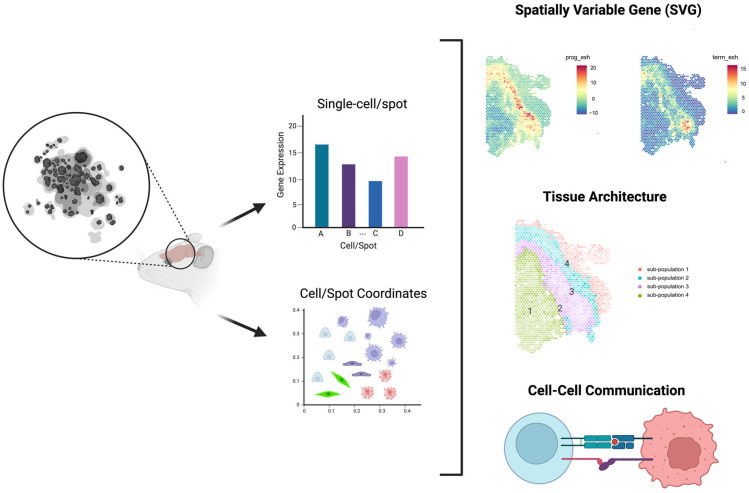
The figure depicts three representative research questions for the analysis of HST data. SVG denotes the identification of a gene with a spatial pattern of gene expression. Tissue architecture refers to the identification of a tissue’s structure through the clustering of similar gene expression patterns. Cell-cell communication, on the other hand, detects the interaction between cells using their spatial information and gene expression data.

**Figure 3 biomolecules-13-00221-f003:**
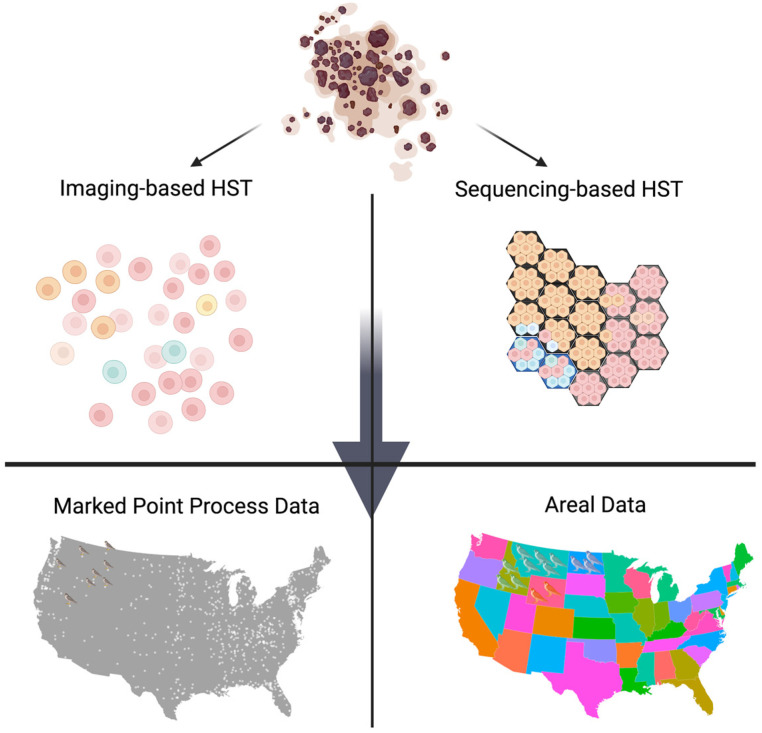
Depending on the type of HST data, it can be considered as either marked point process data or areal data. First, imaging-based HST data can be regarded as marked point process data. For example, cell locations are analogous to the spatial coordinates of birds’ habitats in the US. Its spatial information is modeled through the distance among habitats. Sequencing-based HST data, on the other hand, can be regarded as areal data on a regular grid. Here the spot, which is a group of cells, can be compared to the states’ aggregated bird habitats. Its spatial information is modeled through the adjacency or neighborhood structure.

**Figure 4 biomolecules-13-00221-f004:**
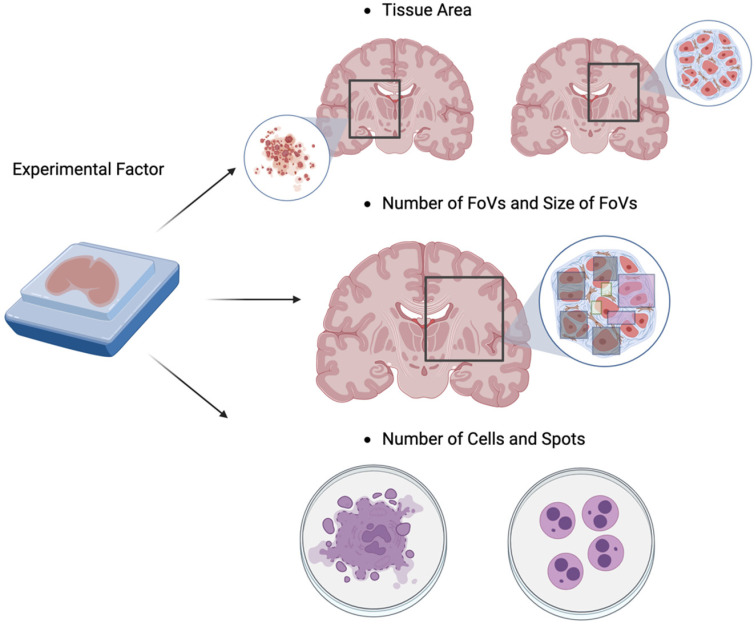
Key experimental factors in designing HST experiments include: (1) the choice of tissue area, (2) the number and sizes of fields of view (FoVs), and (3) the number of cells and spots. These experimental factors can affect the statistical power needed to achieve the research goals, e.g., those mentioned in Figure 2. For example, the choice of tissue area, along with the number and sizes of FoVs, can determine the degree to which the biological aspects of our interest (e.g., interesting cell subpopulations, or cell-cell communications) are captured in the generated HST data. Likewise, the number of cells and spots can affect the signal-to-noise ratios (effect sizes) of the generated HST data.

**Table 1 biomolecules-13-00221-t001:** A table shows six software tools for statistical power analysis for bulk RNA-seq experiments. Each tool is presented along with the citation and the software environments that have been implemented.

		Tool Name [Citation] (Implementation)	
		Pilot Data	Pilot Data with Stored Data
**Type 1 Error**	**Poisson Log-normal**	-	‘Scotty’ [33] (Web Interface)
	**Negative Binomial**	‘RNASeqPower’ [19] (R package)	-
**FDR**		‘ssizeRNA’ [31] (R package)	‘RnaSeqSampleSize’ [34] (R package)
		‘RNASeqPowerCalculator’ [35] (R package)	‘PROPER’ [32] (R package)

**Table 2 biomolecules-13-00221-t002:** A table with information about different software tools for scRNA-seq power analysis with two distinct detection targets. Experimental Factors: cell number (1), individual number (2), Sequencing depth (3).

Detection Target	# of Samples	Tool Name	Experimental Factor	Software	Model	Power Assessment
Cell sub- population	Single sample	‘SCOPIT’ [37]	(1)	R package & Web application	Multinomial	Analytical
		‘howmanycells’		Web application	Negative Binomial	
	Multi sample	‘Sensei‘ [38]	(1), (2)		Beta Binomial	
		‘scPOST’ [39]		R package	Linear mixed model	Simulation- based
DEG		‘scPower’ [40]	(1), (2), (3)	R package & Web server	Negative Binomial	Pseudobulk
		‘hierarchicell’ [41]		R package		Simulation- based
	Single sample	‘powsimR’ [42]	(1)			
		‘POWSC’ [43]	(1), (3)		A mixture of zero-inflated Poisson and log-normal Poisson distributions	
		‘scDesign’ [44]			Gamma-Normal mixture model	

## Data Availability

Not applicable.

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
