# Peer review of "Statistical Power Analysis for Designing Bulk, Single-Cell, and Spatial Transcriptomics Experiments: Review, Tutorial, and Perspectives"

_biomolecules, 2023, doi:10.3390/biom13020221_

Round 1

Reviewer 1 Report

This time-sensitive survey paper is well-written. It summarizes power analysis tools in omics study and could help spatial transcriptomics design. All my concerns below are minor. I hope it can improve the paper.

1. Zero-inflation (ZI) is a very important characteristic of bulk RNA-seq, scRNA-seq, and HST data. I suggest the authors discuss how existing work or tools tackle this issue. For example, there are modified versions of edgeR and DESeq2, using a ZINB-based wanted variation extraction (WaVE) strategy to down weight the inflated amount of zeros. 

Van den Berge, Koen, et al. "Observation weights unlock bulk RNA-seq tools for zero inflation and single-cell applications." Genome biology 19.1 (2018): 1-17.

2. The most interesting part of the paper is Section 4: Power analysis for HST experiments. The authors have introduced the power analysis for different types of questions in the other two gene expression profiling technologies. However, they did not follow the same way for HST. I strongly suggest the authors introduce the three HST questions in more detail: SV gene detection, tissue architecture identification, and cell-cell communication, especially the first two. In fact, cell-type deconvolution is another significant question that the author cannot ignore. 

3. For the SV gene detection question, the authors ignored the latest literature, which have much higher performance than SPARK, spatialDE, and Trendsceek, including BOOST-GP that takes zero-inflation into account, and BinSpect and BOOST-MI take takes lattice structure into account.

Li, Qiwei, et al. "Bayesian modeling of spatial molecular profiling data via Gaussian process." Bioinformatics 37.22 (2021): 4129-4136.

Dries, Ruben, et al. "Giotto: a toolbox for integrative analysis and visualization of spatial expression data." Genome biology 22.1 (2021): 1-31.

Jiang, Xi, Guanghua Xiao, and Qiwei Li. "A Bayesian modified Ising model for identifying spatially variable genes from spatial transcriptomics data." Statistics in Medicine 41.23 (2022): 4647-4665.

4. For tissue architecture identification, the authors also ignore the latest literature that integrates imaging data for better result, including SpaGCN.

Hu, Jian, et al. "SpaGCN: Integrating gene expression, spatial location and histology to identify spatial domains and spatially variable genes by graph convolutional network." Nature methods 18.11 (2021): 1342-1351.

5. Figure 3. It is not very appropriate to make this analogy. Imaing-based HST data is not point process data. First, the location itself is not random. Second, given the location, the measured gene expression is random. So I think point process data with continuous marks should be more appropriate. 

6. For imaging-based HST data analysis, we could also consider how many replicates/samples to use. A very recent method demonstrated that using multi-samples can improve tissue structure identification results.

Li, Zheng, and Xiang Zhou. "BASS: multi-scale and multi-sample analysis enables accurate cell type clustering and spatial domain detection in spatial transcriptomic studies." Genome biology 23.1 (2022): 1-35.

Reviewer 2 Report

Statistical power analysis is important when dealing with NGS data.  There are some power analyses for bulk NGS data. However, power analysis has to be defined well for new types of data including single cell RNAseq and spatial transcriptomics. The review provides an overview about the current efforts for power analysis for scRNAseq and spatial transcriptomics and it describes well about the limitation of current approaches.  The review is very timely and provides detailed background about each approach.  The review is well written. 

Reviewer 3 Report

A very well-written review, highly expected in the field; therefore this submission should be accepted without major concerns. I have noticed only small editorial errors:

1. descriptions on the Fig. 1 should be presented with higher magnitude, at present are highly illegible;

2. in all manuscript: use 'replicas' rather than 'replicates';

3. line 71: after 'Pollen' remove comma;

4. line 77: change 'can' to 'should';

5. consider writing 'p' (for p-value) in italics in all manuscript;

6. line 149: the expansion of 'FDR' abbreviation should be presented on p. 3, line 135 at first;

7. Tables 1 & 2: table headings (e.g. Tool Names, Pilot Data, etc.) should be highlighted by bold;

8. line 229: better: 'cell number', 'individual number';

9. line 293: replace 'works' by 'approaches';

10. line 448 & 456: provide quotation marks for all tools described here;

11. line 480, 523, etc.: delete comma after 'e.g.' (and in all manuscript);

12. line 641: J Abnorm Psychol (in italics);

13. line 643: Curr Dir Psychol Sci (in italics);

14. line 644: Conserv Biol (in italics);

15. line 652: Ann Rev Biomed Data Sci (in italics);

16. line 746, 747: refs. 62, 63: issue number, page range are missing.
